# *Bacillus licheniformis* FA6 Affects Zebrafish Lipid Metabolism through Promoting Acetyl-CoA Synthesis and Inhibiting β-Oxidation

**DOI:** 10.3390/ijms24010673

**Published:** 2022-12-30

**Authors:** Sijia Chen, Weidong Ye, Kendall D. Clements, Ziye Zan, Weishan Zhao, Hong Zou, Guitang Wang, Shangong Wu

**Affiliations:** 1Key Laboratory of Aquaculture Disease Control, Ministry of Agriculture, State Key Laboratory of Freshwater Ecology and Biotechnology, Institute of Hydrobiology, Chinese Academy of Sciences, Wuhan 430072, China; 2University of Chinese Academy of Sciences, Beijing 100049, China; 3School of Biological Sciences, University of Auckland, Private Bag 92019, Auckland 1024, New Zealand

**Keywords:** *Bacillus licheniformis*, lipid metabolism, gut microbiota, acetyl-CoA, β-oxidation

## Abstract

The intestinal microbiota contributes to energy metabolism, but the molecular mechanisms involved remain less clear. Bacteria of the genus *Bacillus* regulate lipid metabolism in the host and are thus commonly used as beneficial probiotic supplements. In the present study, *Bacillus licheniformis* FA6 was selected to assess its role in modulating lipid metabolism of zebrafish (*Danio rerio*). Combining 16S rRNA high-throughput sequencing, micro-CT scan, metabolic parameters measurement, and gene expression analysis, we demonstrated that *B. licheniformis* FA6 changed the gut microbiota composition of zebrafish and increased both the Firmicutes/Bacteroidetes ratio and lipid accumulation. In terms of metabolites, *B. licheniformis* FA6 appeared to promote acetate production, which increased acetyl-CoA levels and promoted lipid synthesis in the liver. In contrast, addition of *B. licheniformis* lowered carnitine levels, which in turn reduced fatty acid oxidation in the liver. At a molecular level, *B. licheniformis* FA6 upregulated key genes regulating de novo fatty acid synthesis and downregulated genes encoding key rate-limiting enzymes of fatty acid β-oxidation, thereby promoting lipid synthesis and reducing fatty acid oxidation. Generally, our results reveal that *B. licheniformis* FA6 promotes lipid accumulation in zebrafish through improving lipid synthesis and reducing β-oxidation.

## 1. Introduction

The gastrointestinal tract of animals is a complex ecosystem that commonly harbors a diverse bacterial community [1] which co-evolves with the host to perform many functions that the host cannot perform on its own [2]. Regulating host energy metabolism [3], including lipid metabolism [4], is an important role of gut microbiota. Lipid is an important nutrient that plays a significant role in the growth, development, metabolism, and other life activities of animals [5]. Recent work has shown that gut microbiota regulate fat storage in mice [6]. Similarly, Semova et al. [7] showed that microbiota stimulate fatty acid uptake and lipid droplet formation in the intestinal epithelium and liver of zebrafish (*Danio rerio*; Danionidae). In addition, the relative abundance of certain bacterial taxa (e.g., the Firmicutes/Bacteroidetes (F/B) ratio) has been demonstrated to be associated with obesity in mammals, which may be due to the increased ability of the ‘obese gut microbiota’ to salvage energy from the diet [8]. More specifically, *Citrobacter* sp. increases lipid accumulation in mesenteric adipose tissue, accompanied by increased total triglyceride (TG) absorption efficiency and TG re-esterification in Nile tilapia (*Oreochromis niloticus*) [9]. Despite the importance of gut microbiota in host lipid metabolism, most work to date focuses on the role of the intestinal microbiome in causing obesity. The role of intestinal microbiota in regulating lipid metabolism remains poorly understood.

The liver is the main site of lipid synthesis and one of the main host organs involved in the regulation of host lipid metabolism by gut microbiota [10]. Numerous studies have shown that some metabolites of gut microbiota, such as short-chain fatty acids (SCFA), can (a) affect the expression of several key enzymes, including acetyl-CoA carboxylase, fatty acid synthase, carbohydrate responsive element binding protein, and sterol responsive element binding protein 1c, and (b) further regulate de novo lipid synthesis in the host liver [6,11]. The SCFA acetate is transported to the liver to generate acetyl-CoA through activity of acyl-CoA synthase short-chain family member 2, and acetate is also an important source for lipogenesis in the liver [12,13]. Acetyl-CoA is a substrate for triglyceride and cholesterol synthesis, and higher concentrations of intracellular acetyl-CoA are more conducive to lipid synthesis [14]. In addition, gut microbiota also metabolize carnitine and influence carnitine levels in the host. This directly influences the process of fatty acids entering mitochondria through carnitine palmitoyltransferase I for β-oxidation [15]. In general, these studies have greatly improved our understanding of how gut microbiota regulate host lipid metabolism. However, our present understanding is largely derived from work on mammals, and the mechanisms by which gut microbiota regulate lipid metabolism in lower vertebrates remain unknown.

*Bacillus* spp. is one of the most commonly used probiotics [16,17], and can promote lipid accumulation in the host. For example, *Bacillus subtilis* can promote the synthesis and transport of triacylglycerol, as well as the digestion and absorption of fat [17]. *Bacillus licheniformis* can promote the growth of common carp (*Cyprinus carpio* L.), change the composition of gut microbiota, and significantly increase the abundance of Firmicutes [18]. On the other hand, *Bacillus* spp. can also reduce host fat accumulation. For example, *Bacillus clausii* decreased TG levels and body lipid deposition in Japanese flounder (*Paralichthys olivaceus*) [19]. In Asian sea bass (*Lates calcarifer Bloch*), addition of *Bacillus licheniformis* and *Bacillus subtilis* resulted in elevated lipase levels and a significant reduction in overall body lipid levels [20]. However, the mechanisms through which *Bacillus* spp. affect lipid metabolism are unclear.

Teleosts are the most taxonomically and ecologically diverse clade of vertebrates [1]. As a model organism, zebrafish have been extensively used to study lipid metabolism [21]. In this study, we added *Bacillus licheniformis* FA6 to zebrafish diet to elucidate the mechanisms by which gut microbiota regulate fat metabolism in lower vertebrates through micro-CT and characterizations of the microbiome, metabolome, and transcriptome.

## 2. Results

### 2.1. Growth Performance

The prepared bacterial solution was added to zebrafish basal diet, and two experimental groups were set up: a low-dose (LD) group (1 × 10^7^ cfu/g) and a high-dose (HD) group (1 × 10^9^ cfu/g). Our growth performance indicators revealed that the weight gain rate (WGR), specific growth rate (SGR), and condition factor (CF) of the HD group were significantly higher than those of control and LD groups, while a significant reduction in feed conversion ratio (FCR) was recorded in the HD group (Figure 1). The WGR, SGR, and CF of the LD group increased compared with the control group, but there were no significant differences (Figure 1). In addition, there was no significant difference in SR between the *B. licheniformis* supplementation treatments and the control group (Figure 1).

### 2.2. Body Fat and Liver Biochemical Indexes

We used a 3D micro-CT scanner to reconstruct 3D images corresponding to areas of zebrafish body fat (Figure 2A). The percentage of body fat in the control, LD, and HD groups were 7.84%, 8.30%, and 10.43%, respectively. Further statistical analysis indicated that the body fat area of the HD group was significantly larger than that of the control group, and the body fat area of the LD group was larger than that of the control group, but not significantly (Figure 2B). Meanwhile, compared with the control group, total triglyceride (TG) increased in the HD group, but the difference was not significant (*p* = 0.207) (Figure 2C). Total cholesterol (TC), high-density lipoprotein cholesterol (HDLC), and low-density lipoprotein cholesterol (LDLC) in liver were significantly higher in the HD group than those in the control group (Figure 2C,D).

### 2.3. Changes in the Gut Microbiota

We determined gut microbiota community composition in control, LD, and HD zebrafish. At the phylum level, the proportion of Firmicutes in the HD group was significantly higher than those in control and LD groups (Figure 3A), and the value of F/B increased significantly (4.6 times of the control group). Proteobacteria accounted for the highest proportion in control and LD groups (51.97% and 57.82%, respectively), while Firmicutes accounted for the highest proportion in the HD group (47.52%). *Bacillus* spp. Was the most dominant genus (36.83%) in the HD group, where it makes up a large proportion of Firmicutes (Figure 3B). However, *Bacillus* spp. Were less prevalent in the control and LD groups (0.26% and 0.73%, respectively) (Appendix A). The Shannon, Simpson, and Pielou_e indexes in the HD group were significantly lower than those in the LD and control groups, and these indexes in the LD group were also lower than those in the control group, but not significantly (Appendix A). Principal coordinate analysis (PCoA) analysis based on Bray–Curtis similarity revealed that samples from control and LD groups clustered together, while samples from the HD group clustered separately (Figure 3C). Further PerMANOVA analysis revealed a significant difference (F = 3.07, *p* = 0.001) in the composition of bacterial communities among control, LD and HD groups, but not between control and LD groups (F = 1.24, *p* = 0.182) (Appendix A).

### 2.4. Variation in Gut Metabolites

As HD and control groups differed significantly in bacterial composition, a metabolomic analysis was performed on these two groups. An orthogonal partial least square-discriminate analysis (OPLS-DA) score plot distinguished samples from control and HD groups (Figure 4A). A total of 81 differential metabolites was detected between the control and HD groups, including 52 upregulated and 29 downregulated differential metabolites (Figure 4B and Appendix A). According to the primary classification, the three main differential metabolites were amino acids and their metabolites (AAMs, 34.6%), fatty acyls (FAs, 21.0%), and organic acids and their derivatives (OADs, 14.8%) (Figure 4B). Most of the AAM differential metabolites, such as glutaconic acid, glutathione reduced form, and TRP-Glu, were upregulated (92.9%). Among the 17 differential FA metabolites, 11 carnitine metabolites, including carnitine C7, C9, C12, C14, C15, C16, and C18, were downregulated (Figure 4B). Most of the OAD differential metabolites, such as sulfoacetic acid, 4-acetylbutyric acid and hydroxyphenyllactic acid, were upregulated (66.7%). Further, Kyoto Encyclopedia of Genes and Genomes (KEGG) enrichment analysis indicated that these differential metabolites were related to biliary secretion, neuroactive ligand–receptor interaction, thyroid hormone signaling pathways, steroid hormone biosynthesis, thyroid hormone synthesis, tryptophan metabolism, arachidonic acid metabolism, and other pathways (Figure 4C).

In addition, some metabolites were highly variable. For example, arachidonic acid ethanolamide, 7α, 25-dihydroxycholesterol, methylmaleic acid and glutamate were almost undetectable in the control group, but highly elevated in the HD group (Appendix A). In contrast, lysophosphatidylserine (16:2/0:0), thromboxane B2, and carnitine C15: 1: DC were highly elevated in the control group, but hardly detected in the HD group (Appendix A). Most of the AAM differential metabolites were positively correlated with each other, and negatively correlated with α-CEHC (Appendix A). Most of the carnitine differential metabolites were positively correlated with each other, and negatively correlated with protocatechuic aldehyde, eicosanoyl-EA, and hydroxyphenyllactic acid (Appendix A).

### 2.5. Multimodal Responses in Lipid Metabolism-Associated Genes

We sequenced transcriptomes from liver tissues of zebrafish from control and HD groups to infer their gene expression profiles. A total of 3154 differentially expressed genes (DEGs), including 1754 upregulated DEGs and 1400 downregulated DEGs, were identified by differential expression analysis (Appendix A). Gene Ontology (GO) and KEGG enrichment analysis revealed that these DEGs were significantly enriched in translation, transcription, fat metabolism, proteasome, PPAR signaling pathway, carbohydrate metabolism, and peroxisomal pathways (Figure 5A,B). Of these, five pathways related to fat metabolism, i.e., steroid biosynthesis, fatty acid biosynthesis, glycerolipid metabolism, glycerophospholipid metabolism, and steroid hormone biosynthesis. Steroid biosynthesis was the most significantly enriched pathway (Figure 5B), and most of the DEGs in this pathway were upregulated genes, such as *srebf2* (sterol regulatory element binding transcription factor 2). In addition, in the fatty acid metabolism pathway, the key genes related to fatty acid synthesis were upregulated, such as *fasn* (fatty acid synthase), *acaca* (acetyl-CoA carboxylase I), and *acacb* (acetyl-CoA carboxylase II) (Figure 5C), while the key genes related to fatty acid oxidation were downregulated, such as *acox1* (acyl-CoA oxidase 1), *cpt1ab* (carnitine palmitoyltransferase 1Ab), and *cpt2* (carnitine palmitoyltransferase II) (Figure 5C).

### 2.6. Validation of DEGs by RT-qPCR

We selected 10 DEGs related to lipid metabolism to further verify whether their expression levels were consistent with RNA-seq. We found that the changes of these genes were consistent with their expression trend of RNA-seq (Figure 6). *Acaca*, *angptl4*, *fasn*, *hmgcra*, *srebf2*, *hmgcs1*, and *acacb* displayed a trend of upregulation, while *cpt1ab*, *acox1*, and *cpt2* displayed a trend of downregulation.

## 3. Discussion

Gut microbiota play an important role in lipid metabolism [22]. However, the mechanisms by which intestinal microbiota regulate lipid metabolism remain poorly understood. Our study revealed that the common intestinal probiotic *B. licheniformis* FA6 [23,24] affected gut metabolites, further influenced the expression of genes related to fat metabolism, and finally involved in zebrafish lipid metabolism. 

The F/B ratio of intestinal microbiota reflects the degree of obesity in the host to a certain extent [8], and a higher F/B ratio increases energy salvage and is more likely to contribute to obesity [25]. In this study, the composition of bacterial communities differed significantly between control and HD groups. Zebrafish fed a high dose of *B. licheniformis* FA6 displayed a significantly higher F/B ratio compared with the control group. Furthermore, micro-CT scans of trunk fat indicated that the body fat area of the HD group was significantly larger than that of the control group. In addition, the TG, TC, HDLC and LDLC in liver of the HD group were significantly higher than those in the control group. Generally, these results suggest that HD addition of *B. licheniformis* FA6 changed the gut microbiota composition of zebrafish and increased the F/B ratio, resulting in lipid accumulation.

An important influence of gut microbiota on host energy metabolism is the conversion indigestible dietary polysaccharides into SCFA, mainly acetate, propionate and butyrate [26]. The predominant SCFA is acetate, which is absorbed into the liver via the blood and may be converted by acetyl-CoA synthetase 2 (ACSS2) into acetyl-CoA for lipid synthesis in the liver [12]. Fernandes et al. [27] show that the F/B ratio is positively correlated with acetate concentration. Here, although we did not measure acetate concentration in the hindgut, we found that the F/B ratio was significantly higher in the HD group, and the associated differential metabolites, such as sulfoacetic acid, were upregulated. Acetate is a primary substrate for lipogenesis, so anything that increases acetate production is likely to increase lipogenesis [13,28]. Notably, *acss2l* (encoding ACSS2) was upregulated in the HD group, suggesting that more acetate was assimilated from the hindgut and converted into acetyl-CoA in this group. Another major source of acetyl-CoA is through pyruvate conversion [29], and the key gene *dlat* (encoding dihydrolipoamide S-acetyltransferase) [30] was downregulated in the HD group, suggesting that the amount of acetyl-CoA converted from pyruvate was lower in the HD group than that in the control group. In short, these results suggest that acetate serves as an important source of acetyl-CoA [31] and promotes lipid synthesis in the liver. This may also be an important reason why the HD group accumulated more lipids than the control group.

Among the differential metabolites in intestinal contents, carnitine was downregulated in the HD group. Gut microbes, including *Bacillus* spp., have the capacity to degrade carnitine [15,32]. Carnitine plays an important role in the β-oxidation of long chain fatty acids [33], influencing the rate of fatty acid oxidation, basal metabolic rate and utilization of fat energy [34]. In humans and other mammals, carnitine deficiency causes severe hyperlipidemia and systemic metabolic syndrome [35,36]. Dietary carnitine supplementation in zebrafish and *Labeo rohita* increases body carnitine concentrations and decreases lipid content in the liver and body [37,38]. The moderately reduced carnitine content in zebrafish intestine in the HD group in the present study might reduce fatty acid oxidation level and basal metabolic rate, and thereby promote lipid accumulation as shown by higher body fat and higher TG levels in the liver in the HD group. Therefore, we hypothesize that *B. licheniformis* FA6 lowers the carnitine levels in zebrafish, which reduced fatty acid oxidation and ultimately led to increased lipid accumulation. However, the specific mechanism needs to be further elucidated.

*Acaca* and *fasn*, two key genes regulating de novo fatty acid synthesis [39], were upregulated in the HD group, indicating that de novo fatty acid synthesis in the liver increased. This finding is consistent with that reported by Zheng et al. [40] and Bäckhed et al. [6], who found that gut microbiota may regulate lipid synthesis in host liver by affecting the expression of some key genes. Notably, we found that *srebf2*, a key intracellular gene regulating cholesterol synthesis, was upregulated in the HD group, consistent with higher total cholesterol content in the HD group. In addition, we found that *acacb* was upregulated in the HD group, which may inhibit the expression of *cpt1ab* [41]. *cpt1ab* is a key rate-limiting enzyme of fatty acid β-oxidation [42]. In fact, we found that *cpt1ab* was downregulated in the HD group, and that the fatty acid oxidation level of zebrafish in the HD group was lower. Generally, these results suggest that *B. licheniformis* FA6 promotes lipid synthesis and reduces fatty acid oxidation, thereby promoting lipid accumulation in zebrafish.

## 4. Materials and Methods

### 4.1. Experimental Bacterial Strain and Feed Preparation

The *B. licheniformis* FA6 strain used in this study was previously isolated and preserved in our laboratory [43]. The strain was inoculated into Luria–Bertani liquid medium (Fisher Scientific, Ottawa, Ontario, Canada) and grown in a shaking incubator at 37 °C for 24 h. The bacterial solution was centrifuged at 2000× *g* for 10 min to harvest the bacterial pellet, washed three times with sterile phosphate buffer saline (pH 7.4), and diluted with sterile water to the desired concentration of 1 × 10^10^ cfu/mL. The composition of the basal diet (control group) is shown in Appendix A. The prepared bacterial solution was added to zebrafish basal diet, and two experimental groups were set up, including low-dose (LD) group (1 × 10^7^ cfu/g) and high-dose (HD) group (1 × 10^9^ cfu/g). All diets were prepared in the Institute of Hydrobiology, Chinese Academy of Sciences, Wuhan, China, and stored at 4 °C until use.

### 4.2. Feeding Experiment and Sample Collection

All animal-handling procedures and experiments were reviewed and approved by the animal ethics committee of the Institute of Hydrobiology, Chinese Academy of Sciences.

Two-and-a-half-month--old AB wild-type zebrafish were purchased from the China Zebrafish Resource Center. Prior to the experiment, the fish were acclimated in 1000 L tanks for two weeks. Following acclimation, 360 fish were randomly placed in nine 15 L aquaria (40 fish per tank), and divided into three groups (3 replicates/group): an LD treatment group, an HD treatment group, and a control. In the feeding experiment, each group was fed to apparent satiation twice daily at 9:00 and 17:00 for six weeks, with 1/3 of the water changed daily. The environment of the feeding system was: water temperature (28 ± 0.5 °C), 14/10 light cycle, pH (7.2 ± 0.2), and dissolved oxygen (6.55 ± 0.31 mg/L). The fish were weighed at the beginning of the experiment. After six weeks feeding, fish in each group were measured and weighed after 24 h fasting. Then, 22 fish from each tank were anesthetized with tricaine methane sulfonate (MS-222, Sigma, St. Louis, MO, USA) at the concentration of 100 mg/L. Due to the small size of zebrafish, intestine and liver samples were pooled from multiple fish for analysis. Intestine and liver samples from six fish were each pooled into two replicate samples (three fish per sample) for microbial diversity analysis and biochemical analysis. Intestine samples from ten fish were pooled into two replicate samples (five fish per sample) for metabolomics analysis, and liver samples from six fish were pooled into two replicate samples (three fish per sample) for transcriptome analysis. All samples were stored at −80 °C until analysis. In addition, two fish per tank were taken after a 24 h fast and anesthetized with MS-222 for body fat analysis.

### 4.3. Growth Performance Measurements

After the feeding experiment, the following growth performance indicators were calculated: initial body weight (IBW), final body weight (FBW), weight gain rate (WGR), specific growth rate (SGR), feed conversion ratio (FCR), condition factor (CF), and survival rate (SR):WGR (%) = 100 × (FBW − IBW)/IBW,(1)
SGR (%/d) = 100 × (ln FBW − ln IBW)/days,(2)
FCR = Total feed intake/Total weight gain,(3)
CF (%) = 100 × FBW/Final length^3^,(4)
SR (%) = 100 × (Initial fish number − Dead fish number)/Initial fish number,(5)

### 4.4. Micro-CT Scan of Zebrafish Body Fat

The fish body was fixed according to the operation instructions of the instrument, and then a 3D Micro-CT (SkyScan 1276 Micro-CT Scanner, Edinburgh, UK) was used to detect the distribution and content of adipose tissue in the fish body following the method of Wang et al. [44].

### 4.5. Determination of Liver Biochemical Indexes

The liver tissue was accurately weighed and homogenized by adding ice-cold normal saline at 1:10. The homogenate was further centrifuged at 4000 r/min for 10 min at 4 °C, then the supernatant was taken and stored at −80 °C until use. The contents of total triglyceride (TG), total cholesterol (TC), high-density lipoprotein cholesterol (HDLC), and low-density lipoprotein cholesterol (LDLC) were determined using commercial kits (Nanjing Jiancheng Bioengineering Institute, Nanjing, China). Total protein concentration was determined using the Detergent Compatible Bradford Protein Assay Kit (Beyotime Institute of Biotechnology, Shanghai, China).

### 4.6. 16S rRNA Sequencing and Analysis

Bacterial genomic DNA was extracted using the OMEGA Soil DNA Kit (M5635-02) (OMEGA Bio-Tek, Norcross, GA, USA). PCR amplification of the bacterial 16S rRNA gene V3-V4 region was performed using the forward primer 338F (5′-ACTCCTACGGGAGGCAGCA-3′) and the reverse primer 806R (5′-GGACTACHVGGGTWTCTAAT-3′). Sample-specific 7 nt barcodes were incorporated into the primers for multiplex sequencing. PCR amplicons were purified with Vazyme VAHTSTM DNA Clean Beads (Vazyme, Nanjing, China) and quantified using the Quant-iT PicoGreen dsDNA Assay Kit (Invitrogen, Carlsbad, CA, USA). After purification and quantification of PCR products, paired-end sequencing was carried out using 2 × 250 bp paired-end sequencing on an Illumina MiSeq platform (Illumina, San Diego, CA, USA). The QIIME2 pipeline (version 2020.2) was used for sequencing data analysis [45]. In addition, DADA2 was used to preprocess and filter reads. Finally, the q2-classifier plugin was used for taxonomic assignment, and Greengenes (version 13_8) was used as the reference sequence database.

### 4.7. Metabolomics Analysis

Metabolites of gut were extracted using conventional methods [46]. Briefly, the samples collected were frozen immediately in liquid nitrogen and then preserved at −80 °C. For metabolite extraction, the samples were thawed slowly on ice. Then, 20 mg sample was homogenized with 400 mL of ice-cold methanol/water (70%, *v*/*v*). The mixed samples were then vortexed for 5 min and centrifuged at 12,000 r/min at 4 °C for 10 min. The centrifuged supernatant was pipetted into a new 1.5 mL Eppendorf tube and centrifuged at 12,000 r/min at 4 °C for another 5 min. After centrifugation, the supernatant was extracted and analyzed by LC-ESI-MS/MS (AB Sciex 6500). The R package ropls was used for orthogonal partial least square-discriminant analysis (OPLS-DA). Variable importance in projection (VIP) values were extracted from the OPLS-DA result, and score plots and permutation plots were generated using R package MetaboAnalystR. The data were log transformed (log2) and mean centering conducted before OPLS-DA. In order to avoid overfitting, a permutation test (200 permutations) was performed. Metabolites that were significantly regulated between groups were determined by VIP ≥ 1 and absolute Log_2_ (Fold Change) ≥ 1. Identified metabolites were annotated using the KEGG Compound database (http://www.kegg.jp/kegg/compound/ (accessed on 3 July 2022)), and annotated metabolites were then mapped to KEGG Pathway database (http://www.kegg.jp/kegg/pathway.html (accessed on 3 July 2022)) [47].

### 4.8. Transcriptome Sequencing and Analysis

Total RNA of liver was isolated using Trizol Reagent (Thermo Fisher Scientific, San Jose, CA, USA). Sequencing libraries were constructed using the TruSeq RNA sample preparation Kit (Illumina, San Diego, CA, USA) following the manufacturer’s protocol. Briefly, purified mRNA was broken into small fragments, the first strand of cDNA was synthesized by reverse transcription with random primers, and then the second strand of cDNA was synthesized. Libraries were purified using the AMPure XP system (Beckman Coulter, Beverly, CA, USA). After purification, library quality was evaluated with the Agilent Bioanalyzer 2100 system, and then sequenced on the Illumina Hiseq4000 platform (2 × 150 bp). HISAT2 (version 2.1.0) was used to map the clean reads to zebrafish reference genomes (GRCz11) [48,49]. StringTie (version 1.3.1c) was used for transcriptional splicing [50]. Htseq (version 0.11) was used to calculate the read counts mapped to each gene as the original expression data of the gene [51]. DESeq2 (version 1.30.1) was used for differential expression analysis [52], and false discovery rate (FDR) < 0.05 was considered as a differential expression gene. ClusterProfiler (version 4.2.0) was used for GO and KEGG enrichment analysis with *p* < 0.05 [53]. Cases where the parameters used are not listed imply that default parameters were used.

### 4.9. RT-qPCR to Verify Gene Expression Levels

A total of 10 differentially expressed genes (DEGs), i.e., *acaca*, *angptl4*, *fasn*, *hmgcra*, *crebf2*, *hmgcs1*, *acacb*, *cpt1ab*, *acox1*, and *cpt2*, were selected as representatives for real-time quantitative PCR (RT-qPCR) verification, and *Rpl13a* was used as the reference gene of mRNA. mRNA was reverse transcribed into cDNA using random primers. RT-qPCR was performed using ChamQ SYBR qPCR Master Mix (Vazyme, Nanjing, China) and the CFX96™ Real-Time PCR Detection System (Bio-Rad, Hercules, CA, USA), with three technical replicates for each sample. Relative gene expression levels were determined using the 2^−ΔΔCT^ method [54]. Primers used in this study are provided in Appendix A.

### 4.10. Statistical Analysis

SPSS software (IBM SPSS 22.0, Chicago, IL, USA) was used for statistical analysis. Data were expressed as mean ± standard deviation (SD). R Package ggplot2 and GraphPad Prism 6.0 software (GraphPad Software, San Diego, CA, USA) were used for plotting. Throughout the study, *p* < 0.05 values are referred to as statistically significant, unless specified otherwise.

## 5. Conclusions

Integrated multi-omics analyses revealed that *Bacillus licheniformis* FA6 affected zebrafish lipid metabolism. At gut microbiota level, *B. licheniformis* FA6 influenced gut microbiota community composition in zebrafish, and increased the F/B ratio, resulting in lipid accumulation. In terms of metabolites, *B. licheniformis* FA6 appeared to promoted acetate production, which increased acetyl-CoA levels and promoted lipids synthesis in the liver. In contrast, addition of *B. licheniformis* lowered carnitine levels, which in turn reduced fatty acid oxidation in the liver. At a molecular level, *B. licheniformis* FA6 upregulated key genes regulating de novo fatty acid synthesis and downregulated key rate-limiting enzyme of fatty acid β-oxidation genes, thereby promoting lipid synthesis and reducing fatty acid oxidation. On the whole, our results suggest that *B. licheniformis* FA6 increased lipid accumulation in zebrafish through promoting lipid synthesis and reducing β-oxidation (Figure 7).

## Figures and Tables

**Figure 1 ijms-24-00673-f001:**
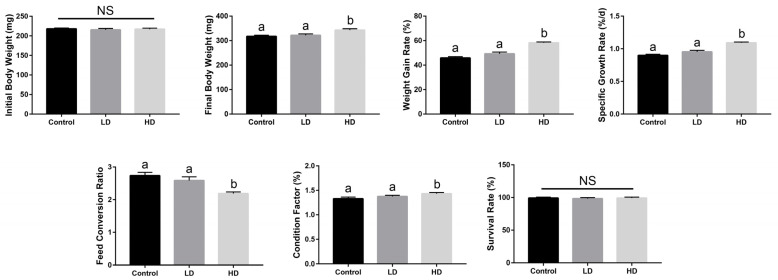
Effects of dietary *Bacillus licheniformis* FA6 on the growth performance of zebrafish. Data are presented as mean ± SD for three replicate groups, with 40 fish in each group. The means of different superscripts in the same row are significantly different. Different letters above the bars indicate significant differences at the 0.05 level (ANOVA and Duncan’s multiple range test), n = 3. “NS” indicates not significant (*p* > 0.05).

**Figure 2 ijms-24-00673-f002:**
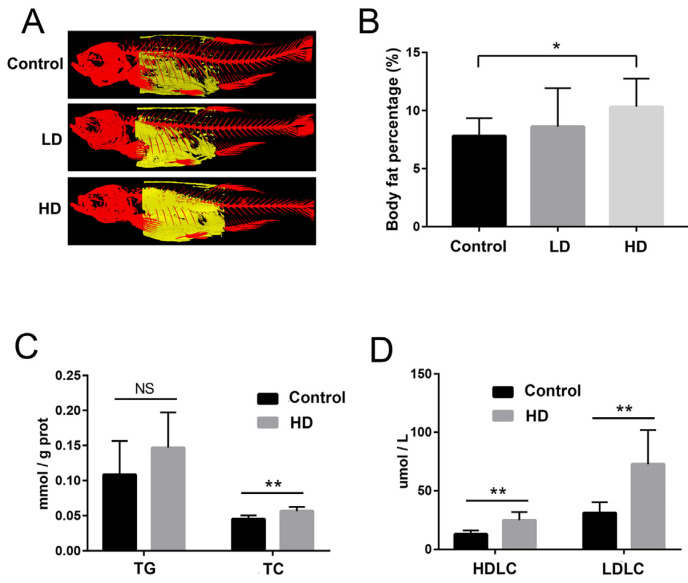
The reconstructed image of body fat revealed by 3D micro-CT, and comparative evaluation of biochemical indexes in zebrafish. (**A**) Typical images of trunk fat (yellow) of fish in each group based on micro-CT. (**B**) Percentage of body fat in each group. (**C**) Bar graph of total triglyceride (TG) and total cholesterol (TC) content in the liver of the control and HD groups. (**D**) Bar graphs of high-density lipoprotein cholesterol (HDLC) and low-density lipoprotein cholesterol (LDLC) levels in the liver of the control and HD groups. * *p*-value < 0.05 compared with the control group. ** *p*-value < 0.01 compared with the control group. “NS” indicates not significant (*p* > 0.05).

**Figure 3 ijms-24-00673-f003:**
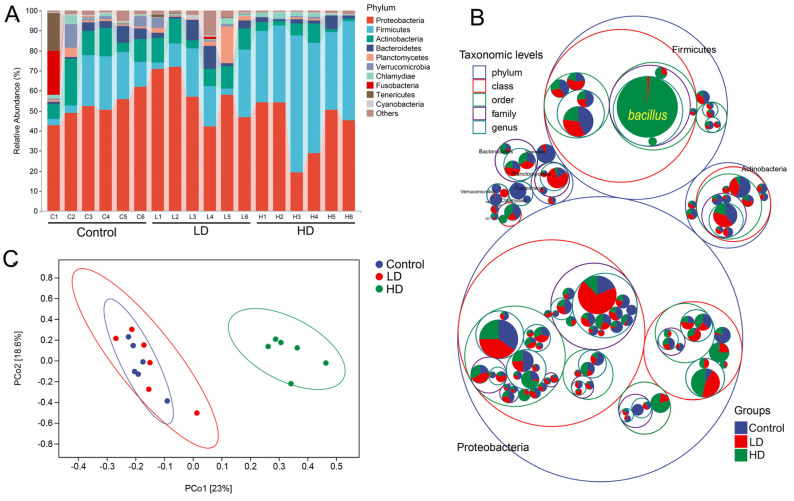
The zebrafish gut microbiota community in the three experimental groups. (**A**) Composition of microbiota communities in the three groups at phylum level. Each bar represents the community of a sample. Only phyla with a mean relative abundance > 1% are shown; low abundance phyla were assigned to ‘others’. (**B**) Microbiota composition at all taxonomic levels in packed circles. The largest circles represent phylum level, and the decreasing circles represent class, order, family and genus. The larger the sector area, the higher the relative abundance of the taxon in the corresponding group. (**C**) Principal coordinate analysis (PCoA) based on weighted UniFrac distances illustrating community dissimilarities across the three experimental groups.

**Figure 4 ijms-24-00673-f004:**
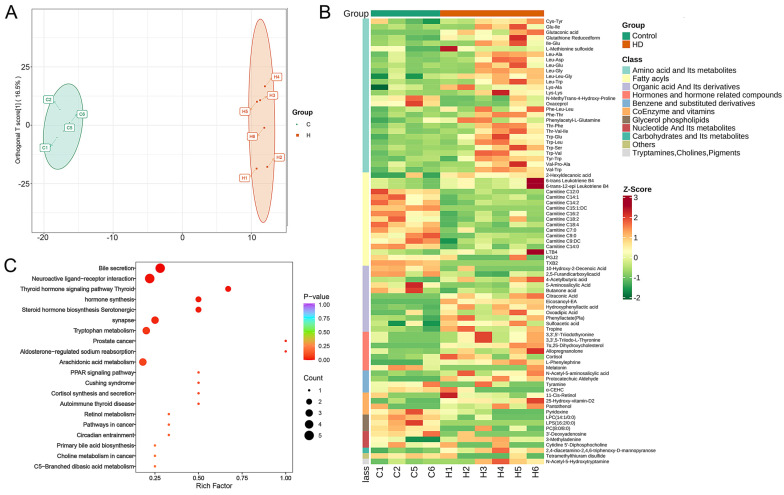
Differences between control and HD groups in gut metabolite composition of zebrafish. (**A**) OPLS-DA score plot of control and HD groups. (**B**) Heat map showing the differential metabolites in different categories, with the leftmost column showing the categories in different colors. (**C**) Bubble diagram of KEGG enrichment results of differential metabolites.

**Figure 5 ijms-24-00673-f005:**
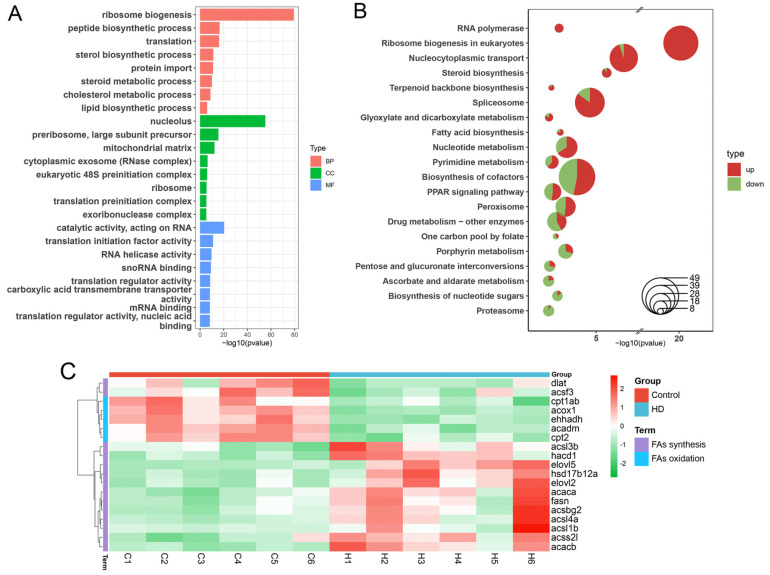
GO and KEGG enrichment analysis of differentially expressed genes (DEGs) in zebrafish liver in control and HD groups. (**A**) GO enrichment analysis of DEGs. After redundant terms are removed from each category, only the top eight terms in *p*-value are displayed. (**B**) KEGG enrichment analysis of DEGs. Red area represents the proportion of upregulated genes in the pathway, and green area represents the proportion of downregulated genes in the pathway. Only the top 15 pathways with *p*-value are shown, and the ordinate order is from largest to smallest according to the proportion of upregulated genes in the pathways. (**C**) Heat map of expression levels of DEGs involved in fatty acid synthesis and oxidation. The key genes related to fatty acid synthesis were upregulated, such as *fasn*, *acaca*, and *acacb*, while the key genes related to fatty acid oxidation were downregulated, such as *acox1*, *cpt1ab*, and *cpt2*.

**Figure 6 ijms-24-00673-f006:**
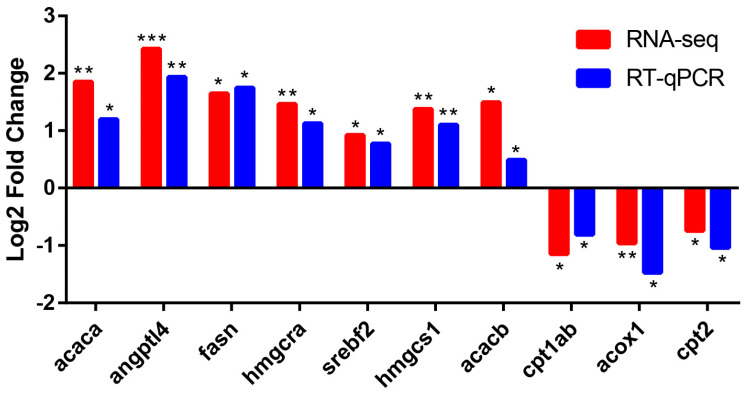
Comparison of RNA-seq data with RT-qPCR data. The *x*-axis shows the names of genes, and the *y*-axis indicates the Log_2_ (Fold Change) with the relative expression (compared to the control group) of each gene. * *p*-value < 0.05 compared with the control group. ** *p*-value < 0.01 compared with the control group. *** *p*-value < 0.001 compared with the control group.

**Figure 7 ijms-24-00673-f007:**
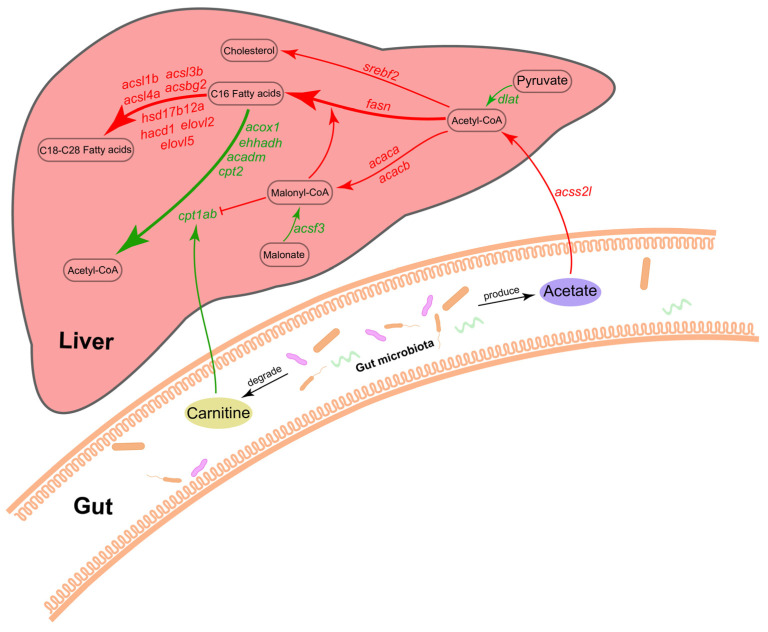
Potential mechanisms for regulation of liver fatty acid metabolism by gut microbiota. Red represents upregulation, and green represents downregulation.

## Data Availability

All clean data files are publicly available at BIG database under the GSA accession number CRA008315 and CRA008314.

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
