# Peer review of "Bacillus licheniformis FA6 Affects Zebrafish Lipid Metabolism through Promoting Acetyl-CoA Synthesis and Inhibiting β-Oxidation"

_ijms, 2022, doi:10.3390/ijms24010673_

Round 1

Reviewer 1 Report

In their manuscript “Bacillus licheniformis FA6 Affects Zebrafish Lipid Metabolism through Promoting Acetyl-CoA Synthesis and Inhibiting β-Oxidation”, the authors describe the effects Bacillus licheniformis supplementation of zebrafish feed has on weight, lipid content and lipid markers of adult zebrafish. The further examine effects on gut metabolites and on liver gene transcripts by metabolomics and transcriptomics. They conclude that supplementation of the diet with this probiotic bacterium increases lipid accumulation via promoting lipid biosynthesis pathways through increased levels of acetate and acetyl-CoA and inhibiting β-oxidation by reduced acyl-carnitine levels and decreases in pathway-related gene expression.

The data presented appear sound, but the presentation should be improved. Some aspects of the experimental design are not explained, experimental and analytical methods are not described in sufficient detail, and many of the results are merely listed, but not discussed or put in to context. Specifically:

1)      At the beginning of the results section, the experimental design is not introduced, and the different types of diet are not defined. The abbreviations for the physiological parameters that were examined are not explained, either.

2)      Why were different types of diversity indices for the gut microbiome examined (Suppl. Fig. 3), and precisely which information was derived from them?

3)      Changes in certain metabolite or gene classes are just listed, but not put into context, and no conclusions of the experiments are presented (except a very general conclusion at the end, which is nearly word by word identical with the abstract). For example, amino acid metabolite changes should be discussed in more detail. How are they related to lipid metabolism?

4)      Conclusions only based on gene expression but not directly supported by metabolite measurements should be clearly stated as such. The same is true if only related metabolites were measured, but not the metabolite discussed. For example, were acetyl-CoA levels directly measured? Levels of free carnitine?

Minor points:

Abstract:

Line 16: replace “assess the role” with “assess its role”

Line 20: replace “promoted” with “promote”

Line 21: replace “lipids” with “lipid”

Line 24: replace “key rate-limiting enzyme of fatty acid β-oxidation genes” with “genes encoding key rate-limiting enzymes of fatty acid β-oxidation”

Similar minor spelling and grammatical errors occur throughout the text and should be corrected.

The conclusion (line 392ff) is nearly word by word identical to the abstract, this should be changed.

Results:

Line 83: Explain the experimental design (HD and LD groups, etc.) first before describing the results. Define all abbreviations on first occurrence in the text, and explain the underlying concepts and the reason why they were examined

Table 1: Show as a graph instead?

Line 121: Shannon, Simpson and Pielou_e indexes; and Bray-Curtis similarity (line 125): Please explain why precisely these were determined and what can be concluded from them.

Line 124: define abbreviation PCoA (Principal Component Analysis)

Line 153 ff: The KEGG enrichment analysis results should be further elaborated: Which metabolites are relevant for neuroactive ligand-receptor interaction? Why is steroid hormone biosynthesis enriched? Why thyroid hormone signaling? Are the changes in these hormonal pathways consistent with previously described systemic effects on lipid metabolism or other metabolic changes?

Line 159 (and elsewhere): “highly expressed”: as this section refers to metabolites and not genes, replace “expressed” by “elevated” or similar.

Line 162ff: How can the negative and positive autocorrelations of metabolites be interpreted in the overall context of metabolic differences under the different diets?

Figure 3B: some rows appear very heterogeneous within treatment groups – discuss. Why are there only 4 control groups? Also, because two samples per tank replicate were generated, samples are not completely independent – how does this affect the statistical analysis?

Generally: print size in the figures is too small. In 3B, also after zooming in on the pdf, metabolite names are not easily legible.

Figure 4: Why are translation and transcription related terms enriched?

Discussion:

Line 219: what does “and finally involved in lipid metabolism” mean?

Line 271f: “In fact, we found that cpt1ab was down-regulated in the HD group, and that the fatty acid oxidation level of zebrafish in the HD group was lower” These statements rest on the observation that mRNA expression of genes involved in fatty acid oxidation and acyl-carnitine levels are lower under the HD diet. A more thorough examination of this issue would require measurement of enzyme activities or flux through the pathway.

Materials and methods:

Line 354: Please provide more details on the mass spectrometry analysis

Line 355: define VIP

Line 372: define FDR

Author Response

Response to Reviewer 1 Comments

Point 1: At the beginning of the results section, the experimental design is not introduced, and the different types of diet are not defined. The abbreviations for the physiological parameters that were examined are not explained, either.

Response 1: We added a brief Experimental design at the beginning of the results, and the more detailed experimental design is in "4.1 Experimental bacterial strain and feed preparation". We added explanations for abbreviations that first appear.

Point 2: Why were different types of diversity indices for the gut microbiome examined (Suppl. Fig. 3), and precisely which information was derived from them?

Response 2: This is a routine microbiome analysis method used to obtain some of the most basic information, which is not directly related to the lipid metabolism subject of this article, but the F/B value information obtained by further microbiome analysis is related to lipid metabolism.

Point 3: Changes in certain metabolite or gene classes are just listed, but not put into context, and no conclusions of the experiments are presented (except a very general conclusion at the end, which is nearly word by word identical with the abstract). For example, amino acid metabolite changes should be discussed in more detail. How are they related to lipid metabolism?

Response 3: Amino acid metabolism is also interesting, but the main focus of this article is lipid metabolism. We don't have a lot of experimental validation that can relate amino acid metabolism or other metabolites to lipid metabolism, so we don't talk much about other metabolites. However, these potential clues are worth further investigation in the future.

Point 4: Conclusions only based on gene expression but not directly supported by metabolite measurements should be clearly stated as such. The same is true if only related metabolites were measured, but not the metabolite discussed. For example, were acetyl-CoA levels directly measured? Levels of free carnitine?

Response 4: Indeed, our conclusions are based on the results of transcriptome and metabolome without experimental verification, and the causal relationship is not strong enough, so our conclusions are all in a weak tone (may, maybe...).

Minor points:

Abstract:

Line 16: replace “assess the role” with “assess its role”

Response:We have modified it according to the reviewer's suggestion.

Line 20: replace “promoted” with “promote”

Response:We have modified it according to the reviewer's suggestion.

Line 21: replace “lipids” with “lipid”

Response:We have modified it according to the reviewer's suggestion.

Line 24: replace “key rate-limiting enzyme of fatty acid β-oxidation genes” with “genes encoding key rate-limiting enzymes of fatty acid β-oxidation”

Response:We have modified it according to the reviewer's suggestion.

Similar minor spelling and grammatical errors occur throughout the text and should be corrected.

Response:We have carefully checked the manuscript and fixed some spelling mistakes.

The conclusion (line 392ff) is nearly word by word identical to the abstract, this should be changed.

 Response:The last sentence of the abstract differs from the last sentence in the conclusion.

Results:

Line 83: Explain the experimental design (HD and LD groups, etc.) first before describing the results. Define all abbreviations on first occurrence in the text, and explain the underlying concepts and the reason why they were examined

Response:We have added explanations for abbreviations when they first appear.

Table 1: Show as a graph instead?

Response:We have modified it according to the reviewer's suggestion.

Line 121: Shannon, Simpson and Pielou_e indexes; and Bray-Curtis similarity (line 125): Please explain why precisely these were determined and what can be concluded from them.

Response: This is a routine microbiome-data analysis method used to obtain basic information. Although these results are not directly related to the lipid metabolism, the F/B value information obtained by further microbiome analysis is related to lipid metabolism.

Line 124: define abbreviation PCoA (Principal Component Analysis)

Response:We have modified it according to the reviewer's suggestion.

Line 153 ff: The KEGG enrichment analysis results should be further elaborated: Which metabolites are relevant for neuroactive ligand-receptor interaction? Why is steroid hormone biosynthesis enriched? Why thyroid hormone signaling? Are the changes in these hormonal pathways consistent with previously described systemic effects on lipid metabolism or other metabolic changes?

Response: The differential metabolites relevant for neuroactive ligand-receptor interaction pathway are tyramine, 3,3',5-Triiodo-L-Thyronine, melatonin LTB4, and cortisol. The differential metabolites relevant for steroid hormone biosynthesis pathway are cortisol and allopregnanolone. The differential metabolites relevant for thyroid hormone signaling are glutathione reducedform and 3,3',5-Triiodo-L-Thyronine. Steroid hormone biosynthesis, thyroid hormone signaling and other pathways were enriched in the present study. However, these pathways are not significantly enriched (Corrected_P-value > 0.05), and have no biological meaning. Therefore, based on these data, we do not know the actual changes occurring in these hormone pathways and their effect on lipid metabolism or other metabolic changes.

Line 159 (and elsewhere): “highly expressed”: as this section refers to metabolites and not genes, replace “expressed” by “elevated” or similar.

Response:We have modified it according to the reviewer's suggestion.

Line 162ff: How can the negative and positive autocorrelations of metabolites be interpreted in the overall context of metabolic differences under the different diets?

Response:In our opinion, the metabolites with positive correlation may be the decomposition products of the same metabolite or multiple metabolites with similar functions, while the negative correlation indicates that the increase of the level of some metabolites may be unfavorable to the synthesis of some metabolites.

Figure 3B: some rows appear very heterogeneous within treatment groups – discuss. Why are there only 4 control groups? Also, because two samples per tank replicate were generated, samples are not completely independent – how does this affect the statistical analysis?

Response:These differential metabolites were determined by statistical tests. Some metabolites, such as tyramine, vary widely from individual to individual, but statistical tests still show that they are differential metabolites. In fact, we initially had six control groups, but PCA analysis showed that two control groups were outliers. We removed the two samples.

Two samples per tank replicate were generated. This is common method used in zebra fish experiment. It increases the number of biological replicates and makes the results of statistical analysis more reliable.

Generally: print size in the figures is too small. In 3B, also after zooming in on the pdf, metabolite names are not easily legible.

Response:In order to enlarge the font size in Figure 3, we show Figure 3 D&E as a Supplementary Material (Figure S4), and enlarge the font size in Figure 3.

Figure 4: Why are translation and transcription related terms enriched?

Response:This result is derived from transcriptome analysis and we did not perform experimental verification. The liver is the core organ of protein synthesis in fish. Therefore, we speculate that the enrichment of translation and transcription in the liver of the HD group implies increased protein synthesis, which may be related to the faster growth rate of zebrafish in the HD group.

Discussion:

Line 219: what does “and finally involved in lipid metabolism” mean?

Response:This means that B. licheniformis FA6 can participate in zebrafish lipid metabolism.

Line 271f: “In fact, we found that cpt1ab was down-regulated in the HD group, and that the fatty acid oxidation level of zebrafish in the HD group was lower” These statements rest on the observation that mRNA expression of genes involved in fatty acid oxidation and acyl-carnitine levels are lower under the HD diet. A more thorough examination of this issue would require measurement of enzyme activities or flux through the pathway.

Response:We strongly agree with the reviewer's statement, and we will further verify these findings in the future.

Materials and methods:

Line 354: Please provide more details on the mass spectrometry analysis

Response:We have modified it according to the reviewer's suggestion.

Line 355: define VIP

Response:We have modified it according to the reviewer's suggestion.

Line 372: define FDR

Response:We have modified it according to the reviewer's suggestion.

Reviewer 2 Report

I consider the article to be very good paper. Below are some comments that may be useful to the authors to improve the manuscript.

Abstract:

It can be mentioned that bacteria representing the genus Bacillus are used as probiotics to emphasize why this one was chosen.

2. Results

Tabele 1.  Explain the meaning of indices a and b in the description.

Figure 3.

Please enlarge the font in figure 3 A, B, C, D because it is illegible even on an A4 printout.

3. Discussion

What I miss in the discussion is a broader consideration of the results described in paragraph 2.1. Growth performance. Is this a standard phenomenon in similar studies with fish as it is with other species? Only one carp study and effect on growth was mentioned in the discussion.

Author Response

Response to Reviewer 2 Comments

Point 1: It can be mentioned that bacteria representing the genus Bacillus are used as probiotics to emphasize why this one was chosen.

Response 1: In accordance with the reviewer's suggestion, we added this to the abstract.

Point 2: Tabele 1.  Explain the meaning of indices and b in the description.

Response2:We have modified it according to the reviewer's suggestion.

Point 3: Please enlarge the font in figure 3 A, B, C, D because it is illegible even on an A4 printout.

Response3:In order to enlarge the font size in Figure 3, we show Figure 3 D&E as a Supplementary Material (Figure S4), and enlarge the font size in Figure 3. Note that we have replaced Table 1 with a picture, so Figure 3 has become Figure 4.

Point 4: What I miss in the discussion is a broader consideration of the results described in paragraph 2.1. Growth performance. Is this a standard phenomenon in similar studies with fish as it is with other species? Only one carp study and effect on growth was mentioned in the discussion.

Response4:Growth is also of great importance, but the subject of our article is lipid metabolism. Continuing to discuss growth may distract from this focus, so we did not discuss growth in the discussion section.
